# Evaluation of a Webinar to Increase Health Professionals’ Knowledge about Myalgic Encephalomyelitis/Chronic Fatigue Syndrome (ME/CFS)

**DOI:** 10.3390/healthcare11152186

**Published:** 2023-08-02

**Authors:** Laura Froehlich, Jasmin Niedrich, Daniel B. R. Hattesohl, Uta Behrends, Claudia Kedor, Johannes-Peter Haas, Michael Stingl, Carmen Scheibenbogen

**Affiliations:** 1CATALPA Center of Advanced Technology for Assisted Learning and Predictive Analytics, FernUniversität in Hagen, 58097 Hagen, Germany; 2Faculty of Psychology, FernUniversität in Hagen, 58097 Hagen, Germany; jasminniedrich@gmail.com; 3German Association for ME/CFS, 20146 Hamburg, Germany; daniel.hattesohl@dg.mecfs.de; 4MRI Chronic Fatigue Center for Young People (MCFC), Children’s Hospital, Technical University of Munich, 80333 München, Germany; uta.behrends@tum.de; 5Institute of Medical Immunology, Charité—Universitätsmedizin Berlin, Corporate Member of Freie Universität Berlin and Humboldt-Universität zu Berlin, 10117 Berlin, Germany; claudia.kedor@charite.de (C.K.); carmen.scheibenbogen@charite.de (C.S.); 6Center for Treatment of Pain in Young People, German Center for Pediatric and Adolescent Rheumatology, 82467 Garmisch-Patenkirchen, Germany; haas.johannes-peter@rheuma-kinderklinik.de; 7Facharztzentrum Votivpark, 1090 Vienna, Austria; ordination@neurostingl.at

**Keywords:** myalgic encephalomyelitis, fatigue syndrome, chronic, ME/CFS, education, medical, continuing, webinars

## Abstract

Myalgic encephalomyelitis/chronic fatigue syndrome (ME/CFS) is a severe chronic illness and patients with ME/CFS are often medically underserved in Germany and other countries. One contributing factor is health professionals’ lack of knowledge about epidemiology, diagnostic criteria, and treatment of ME/CFS. Opportunities are scarce for health professionals to receive continuing medical education on ME/CFS. The current research addressed this need for further education and investigated the gain of knowledge from a webinar for German-speaking health professionals. In two studies (total sample: *N* = 378), participants in the intervention condition completed a knowledge test twice (before and after webinar participation). Study 2 also included a waiting-list control condition with repeated response to the knowledge test without webinar participation between measurements. Results showed that at baseline, most participants had seen patients with ME/CFS, but confidence in diagnosing and treating ME/CFS was only moderate-to-low. In the intervention condition, but not in the control condition, knowledge about ME/CFS increased between the first and the second knowledge test. These results indicate that the webinar was successful in increasing health professionals’ knowledge about ME/CFS. We concluded that webinars can be a cost-efficient and effective tool in providing health professionals with large-scale continuing medical education about ME/CFS.

## 1. Introduction

Myalgic encephalomyelitis/chronic fatigue syndrome (hereafter: ME/CFS) is a debilitating chronic illness of to date unknown etiology. Symptoms include profound exhaustion, muscle weakness and fatigability, pain, sleep disturbances, cognitive dysfunction, orthostatic intolerance, and flu-like symptoms [1,2]. The hallmark symptom is post-exertional malaise, a worsening of symptoms after minimal physical or mental exertion [3,4]. A number of physiological abnormalities have been connected to ME/CFS, including indicators of autoimmunity [5,6] as well as impaired energy metabolism [7,8] and cardiovascular function [9,10]. Although ME/CFS has been officially classified as a neurological illness by the World Health Organization since 1969, it is still largely unrecognized or misunderstood by physicians [11].

Based on a meta-analysis of 46 studies, the pre-pandemic average prevalence of ME/CFS was estimated at 0.39% of the population [12]. In Germany, where the current research was conducted, this would translate to 332,000 individuals (including 54,000 children and adolescents) affected by ME/CFS. ME/CFS is frequently triggered by viral infections, and was demonstrated to overlap with long COVID-19 syndrome [13,14]. Thus, it may be assumed that the COVID-19 pandemic likely will lead to a further increase in ME/CFS prevalence [15,16,17,18].

In several countries, including Germany and Switzerland, patients with ME/CFS are medically underserved in that they encounter obstacles to receiving a diagnosis in a reasonable amount of time and to accessing general and specialized medical care [19,20,21,22]. Patients frequently report being dissatisfied with their medical care and experience stigmatization due to the misconception of ME/CFS being a psychosomatic and/or psychiatric illness [22,23,24]. A major contributing factor to this insufficient and unsatisfying medical care situation of patients with ME/CFS is health professionals’ lack of knowledge about the symptoms, diagnostic criteria, and treatment of ME/CFS [19,21,25]. For example, a systematic review of 33 studies investigating general practitioners’ (GP) knowledge about ME/CFS by Pheby et al. [26] showed that a substantial proportion of GPs did not accept ME/CFS as a genuine clinical entity and even those who did lacked confidence in diagnosing or managing it. Similarly, a survey conducted among 23 experts from the European ME/CFS Research Network (EUROMENE) [11] demonstrated that experts believed that only a small minority of GPs in their country were able to recognize ME/CFS, and were confident in diagnosing and managing it. Moreover, Hng et al. [25] conducted a survey about the knowledge and experience of ME/CFS among 44 UK hospital doctors. Participants reported having very limited formal teaching, but some clinical experience with ME/CFS. Furthermore, 91% falsely believed that ME/CFS was at least in part psychological. Knowledge about the general epidemiology of ME/CFS was high; however, knowledge about diagnostic criteria, treatment, and management was substantially lower. Taken together, these studies showed considerable gaps in the knowledge of health professionals like GPs and hospital doctors. This results in obstacles to patients with ME/CFS receiving an appropriate diagnosis and medical care. The misconception of ME/CFS as an illness based on psychological and/or psychiatric causes bears the risk of recommending potentially harmful activity training, like graded exercise therapy to patients [2,27].

A potential reason for the inadequate knowledge about ME/CFS among health professionals is the scarcity of (under)graduate formal education and Continuing Medical Education (CME) about ME/CFS. Concerning education during medical studies, Jason et al. [28] showed that ME/CFS was underrepresented in US medical textbooks. In Germany, specific diagnostic guidelines for ME/CFS were first published in 2022 [29] and the German National Competence-Based Catalogue (NKLM) for universities and medical schools [30] adheres to incomplete, outdated, and potentially harmful information about ME/CFS. Concerning CME, to date, there are hardly any opportunities for health professionals in German-speaking countries to receive detailed education about the etiology, diagnostics, and treatment of ME/CFS. Hospital and panel doctors in Germany and Austria are required to receive 250 credit points for participation in CME every five years. In both countries, the topics and medical fields to be studied during CME can largely be selected by the doctors (in Germany, 150 of the 250 points need to be collected in one’s specialty field of medicine). Credit points can be awarded for participation in different CME formats, including medical conferences, face-to-face seminars, or virtual web-based training like webinars.

The main objective of the current research was to provide CME about ME/CFS to health professionals in German-speaking countries, which is accessible for a large number of participants, in order to close the knowledge gaps about ME/CFS among health professionals. To achieve this objective, we chose the CME format of a live webinar (web-based synchronous training available for a large number of participants). Webinars are a flexible and cost-effective source of information and education, and have become increasingly frequent especially since the beginning of the COVID-19 pandemic. With webinars, one can take advantage of digitalization by reaching a large number of participants from different regions simultaneously, and spread the knowledge of the few available experts about ME/CFS as widely as possible [31]. Systematic reviews conducted pre-pandemic showed that virtual CME like webinars was equally effective in imparting knowledge to health professionals than traditional face-to-face CME [32,33]. Advantages of virtual CME are higher flexibility, accessibility, convenience, and cost-effectiveness compared to face-to-face CME that often involves traveling and staying overnight away from home. Disadvantages include, for instance, technical barriers like insufficient internet bandwidth, or software and usability problems [32,33,34]. During the pandemic, opportunities to participate in webinars and other virtual forms of CME skyrocketed and are becoming more and more common worldwide. For example, in a representative sample of 2400 hospital and panel doctors in Germany, in 2022, 68% indicated having participated in at least one live webinar as part of their CME, compared to 22% in 2020 [35]. The emerging trend towards online CME seems to persist even after the end of COVID-19 restrictions [34,36]. A recent survey with almost 2000 German physicians showed that 75% wished to participate in live webinars with recordings made available after the webinar [37]. This trend towards webinars as CME was accelerated by the pandemic also in other world regions. Physicians of different specialties in various countries ranging from North America and Asia to Arabic and North African countries reported that during and after the pandemic, they increasingly participated in webinars, were predominantly satisfied with the education they received, and preferred a combination of face-to-face and online CME in the future [38,39,40,41,42]. To meet the needs of health professionals and make use of the recent technological developments, we implemented a CME event about ME/CFS as a live webinar that was recorded and afterwards provided it as an on-demand online CME.

Providing continuing education about ME/CFS for health professionals is a demand recognized on the European level as well [11,25]. However, to date, there is no study demonstrating that participation in continuing medical education increases health professionals’ knowledge about the epidemiology, diagnostic criteria, and treatment of ME/CFS. To close this gap, in the present research, we conducted two studies evaluating the effects of a webinar on health professionals’ knowledge about ME/CFS. We based our research on a survey designed by Hng et al. [25], but employed a repeated-measures design (participants completed the questionnaire before and after webinar attendance). The first aim of the current research was to assess health professionals’ baseline knowledge of and experience with ME/CFS before attending the webinar. The second aim was to evaluate whether participation in the webinar increased participants’ knowledge about the symptoms, diagnostic criteria, and treatment of ME/CFS.

### 1.1. The Present Research

In order to empirically examine health professionals’ knowledge of and experience with ME/CFS, we investigated prior education about ME/CFS, experience and confidence with diagnosing and dealing with patients with ME/CFS, and knowledge about the illness. The research adhered to principles of open science: We pre-registered the hypotheses prior to data collection, and materials, data, and analysis code/ outputs are provided on the Open Science Framework (OSF; https://osf.io/8m32c/, accessed on 25 July 2023). We conducted two separate data collections with German-speaking health professionals who registered for the webinar “Postviral Illnesses: ME/CFS and Long COVID” that took place in October 2021 (Study 1) and September 2022 (Study 2). Study 1 included a pre-post design with two measurement points (before/ after webinar participation). Study 2 employed the same pre-post design for the intervention group and additionally included a waiting-list control group (two measurements before webinar participation). The scientific program of the webinar was organized by the Charité University Medicine Berlin and the Technical University of Munich. The conduction of the webinar with an external provider was funded by the German and Austrian Associations for ME/CFS. It included lectures from experts in research and clinical practice in the field of ME/CFS and was aimed at general health practitioners and specialists from other relevant medical fields.

### 1.2. Hypotheses

The pre-registered hypotheses were very similar in both studies. Study 2 was conducted to substantiate that pre-post differences in knowledge about ME/CFS can be attributed to webinar attendance by supplementing a control group (no webinar attendance between measurements) to the study design. To increase test power, we present combined analyses for both studies where applicable and statistically controlled for sample (Study 1 vs. 2) in additional robustness checks. First, we aimed at replicating previous results about formal teaching, experience, confidence, and knowledge about ME/CFS from Hng et al. [25] in a German-speaking sample of health professionals. Second, we investigated the effects of webinar participation on confidence and knowledge about ME/CFS. We tested the following pre-registered hypotheses (for preregistrations see: https://osf.io/8m32c):
**Hypothesis 1 (H1).** Before webinar attendance, the majority of health professionals have not received formal teaching about ME/CFS (average below 50%; H1a) and do not feel confident in diagnosing patients with ME/CFS (average below 50%, H1b) and dealing with patients with ME/CFS (average below 3.5 on a scale from 1 to 7, H1c).
**Hypothesis 2 (H2).** In the intervention group, knowledge about ME/CFS is higher at T2 compared to T1.
**Hypothesis 3 (H3).** (Study 2 only): There are no differences between the control group and the intervention group concerning H1 (baseline knowledge).
**Hypothesis 4 (H4).** (Study 2 only): In the control group, the knowledge increase between T1 and T2 is smaller than in the intervention group or non-significant. (In Study 1, we pre-registered separate hypotheses for different knowledge domains (i.e., general epidemiological characteristics, definitions and clinical understanding, diagnostic process and diagnostic criteria, treatment) of the survey as presented in Hng et al. [25]. However, internal consistency analyses showed that items could not be reliably aggregated to subscales reflecting these knowledge domains (0.32 < Chronbach’s α < 0.68). Therefore, we aggregated all items to an overall knowledge score, which showed sufficient internal consistency.).

## 2. Materials and Methods

### 2.1. Procedure

Language for the webinar and all study materials was German. Study 1 was conducted between October and November 2021, and Study 2 was conducted in September 2022. The webinars were advertised via newsletters from patient organizations and the Charité University Medicine. The maximum capacity of 1000 participants was reached in both webinars (overall, 18.90% of the webinar attendees participated in the evaluation study). Participants who registered for the webinar received personalized email invitations for the first questionnaire (T1) prior to the webinar. In Study 1, all participants were assigned to the intervention condition, and in Study 2, participants were randomly assigned to the intervention condition or the control condition via an automated process implemented in the online survey. Participants in the intervention condition who completed T1 received personalized email invitations to the second questionnaire (T2) after the webinar. Participants in the control condition who completed T1 received personalized email invitations to T2 until three days before the webinar. The live webinar had a duration of 120 min and consisted of talks by expert researchers and clinicians on the topics of epidemiology, diagnostic criteria, as well as clinical and outpatient care of adults and children/ adolescents with ME/CFS and/ or Long-COVID (for content of the webinars, see Table 1). The invited speakers had long-term experience in biomedical research and/or clinical care of ME/CFS, Long-COVID, and Postural Tachycardia Syndrome (POTS). The talks were followed by a Q&A session. Participation was free of charge. Participants from Germany and Austria could receive three educational points from medical associations (Hamburg Medical Association and Austrian Academy of Physicians), for which they additionally had to complete a multiple-choice test. Educational points were awarded independent of participation in the evaluation study. Since January 2023, the webinar of Study 2 is available as an on-demand educational course for physicians (https://www.mecfs.de/was-ist-me-cfs/informationen-fuer-aerztinnen-und-aerzte/on-demand-fortbildung/, accessed on 25 July 2023).

In accordance with the EU General Data Protection Law, the Declaration of Helsinki, as well as standards of the American Psychological Association (APA), and the German Psychological Society (DGPs), participants were informed about the aims, procedure, and duration of the study and provided written consent for participation. Then, they generated a pseudonymized code to match their data from T1 and T2 while keeping their contact information separate from questionnaire responses. Subsequently, they completed the survey adapted from Hng et al. [25] at both measurement points. At T1, they additionally provided demographic information including age, gender, occupation, medical specialty, and country of residence. Finally, at the end of the surveys, participants were again given the opportunity to consent to their data being used for scientific purposes.

### 2.2. Measures

Materials were translated from English to German and adapted to the German health care system by the project team. The complete codebook with materials in German and English is available on the OSF.

#### 2.2.1. Prior Teaching, Experience with ME/CFS, Confidence

Formal teaching was assessed with two items (“I have received formal teaching on ME/CFS during medical school”, Yes/ No; and “I have received formal teaching on ME/CFS during an educational course in my professional life (clinic, medical association, congress, etc.)”, Yes/No). Experience with patients was assessed with one item (“I have seen patients with ME/CFS (in the practice/clinic)”, on a Likert scale ranging from 1 = very often to 5 = never). Confidence about the diagnosis of ME/CFS was assessed with one item (“I know how to diagnose ME/CFS”, Yes/No). Confidence in dealing with patients with ME/CFS was assessed with one item (“I feel confident in dealing with ME/CFS patients”, on a Likert scale ranging from 1 = fully agree to 7 = do not agree at all).

#### 2.2.2. Knowledge about ME/CFS

The knowledge test included 17 items based on Hng et al. [25]. In the original work, the authors divided the test into four knowledge domains: general epidemiological characteristics (7 items), definitions and clinical understanding (4 items), diagnostic process and diagnostic criteria (4 items), as well as treatment (2 items). Two additional items assessed knowledge about disability due to ME/CFS, but are not reported in the current manuscript.

#### 2.2.3. Demographics

Demographic data included year of birth, gender (male/female/non-binary), occupation (physician/other), medical specialty, work context (clinical, private practice, other), and country of residence (Germany, Austria, Switzerland, other).

### 2.3. Participants

A priori power analyses to estimate the minimum required sample sizes for the studies were conducted with the R package superpower. In Study 1, to investigate H2, we estimated a required sample size of *N* = 85 to reach the desired power of 0.95, based on a repeated-measures analysis of variance (ANOVA) with the hypothesized pattern of means and a small-to-medium effect size of *d* = 0.2, *SD* = 0.5, a correlation between the repeated measures of *r* = 0.5 and an α level of 0.05. In Study 2, we estimated the required sample size to test H4 with a two-way repeated-measures ANOVA with the within-participants factor time (T1 vs. T2), the between-participants factor condition (control vs. intervention) and their interaction. Based on the results of Study 1, we assumed an effect size of 3.26 points (*SD* = 3.41) for the increase in the knowledge test in the intervention condition and no increase in the control condition. For a correlation of the repeated measures of *r* = 0.5, an α level of 0.05, the hypothesized interaction effect could be detected with a power of 0.95 in a sample of *n* = 30 participants per group (total *N* = 60) who participated in both measurement points.

The estimated sample sizes were achieved in both studies (Study 1: T1: *n* = 206, T2: *n* = 145; Study 2: T1: *n* = 172, T2: *n* = 128). We applied the following pre-registered exclusion criteria: Participants were excluded if they did not consent to scientific use of their data (Study 1: T2: *n* = 1) and from the analyses pertaining to T1–T2 comparisons if their participant codes could not be matched (Study 1: *n* = 4, Study 2: *n* = 2).

To investigate H1, we combined the baseline samples of both studies, which resulted in a total sample size of *N* = 378 at T1. To investigate H2–H4, we combined the matched T1–T2 samples of both studies, the total sample size of the combined dataset was *N* = 266 (control group: *n* = 50, intervention group: *n* = 216). In the combined baseline sample, 266 (70%) participants were female, 106 (28%) male, and 5 (1%) non-binary (1 missing value). Participants were predominantly from Germany (*n* = 278, 74%) or Austria (*n* = 81, 21%). Further participants were from Switzerland (*n* = 14), and other countries (*n* = 4, 1 missing value). Age ranged between 24 and 85 years (*M* = 49.07, *SD* = 11.01). The majority of participants (87%) were physicians, the remaining participants worked in other health professions (e.g., psychological psychotherapists, neuropsychologists, physiotherapists, nurses). Among physicians, the most frequent medical specialties were internal medicine (*n* = 54), psychiatry (*n* = 33), neurology (*n* = 29), and pediatrics (*n* = 26). One-third of the participants (*n* = 114) reported working in a clinical context, half of the participants (*n* = 192) worked in a private practice, and the remaining participants in other contexts.

### 2.4. Data-Analytical Strategy and Transformations

Data were analyzed using R/R-Studio Version 43. All items were recoded such that higher values reflect higher agreement/knowledge. To investigate H1a–c, we conducted one-proportion *Z*-tests against equal proportions and a one-sample *t*-test against the theoretical midpoint of the scale. To investigate H2, we aggregated the items of the knowledge test to an overall test score and conducted a paired-samples *t*-test and computed a Linear Mixed Model with Time (T1 vs. T2) and Study (1 vs. 2) as factors (In Study 1, pre-post comparisons were preregistered to be analyzed with a 2 × 4 repeated-measures ANOVA with time (T1 vs. T2) and knowledge domain as factors. Since separate scales for the knowledge domains could not be constructed, we conducted a paired-samples *t*-test on the overall test score instead.). To investigate H3 and H4, we computed a Linear Mixed Model with Time (T1 vs. T2) and Condition (Control vs. Intervention) as factors. In exploratory analyses, we compared the means of the single items of the knowledge test between T1 and T2 with paired-samples *t*-tests with Bonferroni corrected *p*-values.

### 2.5. Dropout Analyses

We compared participants that answered both questionnaires (T1 and T2; complete participants, *N* = 266) and those that only answered the first questionnaire or were excluded from analyses pertaining to T2 (T1; dropouts, *N* = 112) concerning demographic variables (age, gender dummy-coded as male vs. female) as well as experience and confidence with ME/CFS and knowledge about ME/CFS at T1. Both groups did not significantly differ from each other (*t*s < 0.1.41, *p*s > 0.161), indicating that there was no selective dropout between measurements. As a further analysis pertaining to risk of bias [43], we tested whether participants in the intervention condition and control condition were equivalent at baseline concerning sociodemographics. There were no differences on gender (*t*(259) = 0.97, *p* = 0.332). However, participants in the control condition were, on average, six years younger than participants in the intervention condition (*t*(264) = 3.59, *p* < 0.001).

## 3. Results

### 3.1. Hypothesis 1: Baseline Formal Teaching and Confidence

Within the combined sample of Studies 1 and 2 (*N* = 378), only a minority of participants (11%) reported that they had never seen a patient with ME/CFS in their clinic or practice. To test Hypotheses 1a–c, we inspected the proportions of formal teaching and confidence in diagnosing ME/CFS as well as the mean level of confidence in dealing with patients with ME/CFS at T1. Only a fraction of participants (10%) reported that they had received formal teaching on ME/CFS during medical studies. However, 44% reported to have received continuing medical education on ME/CFS. Taken together, 49% of participants had received some form of teaching on ME/CFS. Contrary to H1a, a one-sided one-proportion *Z*-test showed that this proportion was not significantly lower than a proportion of 0.50, *χ*^2^(1) = 0.07, *p* = 0.399, 95% CI [0.00; 0.54]. In line with H1b, 43% of participants reported that they know how to diagnose ME/CFS, which was significantly lower than a proportion of 0.50, *χ*^2^(1) = 6.88, *p* = 0.004, 95% CI [0.00; 0.47]. Finally, tentatively in line with H1c, participants indicated slightly lower than medium confidence in dealing with patients with ME/CFS (*M* = 3.34, 95% CI [3.18; 3.50], on a scale from 1 to 7). However, the mean was only marginally significantly different from the midpoint of the scale (3.5), *t*(377) = 1.95, *p* = 0.052.

### 3.2. Hypothesis 2: Knowledge Increase in Intervention Condition

Next, we investigated in the combined sample of participants assigned to the intervention condition (Studies 1 and 2; *n* = 216) whether knowledge about ME/CFS was higher after participation in the webinar compared to before the webinar. To do so, we re-coded the questions in the knowledge test, such that correct responses received one point for each response option and aggregated the responses to a sum score for the complete knowledge test (Cronbach’s α: T1 = 0.73; T2 = 0.64). All items, response options, and correct responses are displayed in Table 2 (left side). The maximum number of achievable points was 36. At T1, participants on average received 27.13 points (range: 16–35 points), and at T2, the average points received were 30.14 (range: 22–35 points). A paired t-test showed that knowledge after webinar participation was significantly higher than baseline knowledge before webinar participation, *t*(215) = 13.65, mean difference = 3.01, 95% CI [2.58; 3.45], *p* < 0.001. As an additional robustness check, we investigated potential differences in knowledge increase between Study 1 and Study 2 in a Linear Mixed Model with Time (T1, T2) and Study (Study1, Study2) as predictors. Predictors were entered as fixed effects and we included a random intercept for participants to account for the repeated measurement of the knowledge test. Neither the main effect of Study was significant (F(1, 216) = 3.12, *p* = 0.079) nor the interaction of Time and Study (F(1, 216) = 2.27, *p* = 0.133). However, the main effect of Time remained significant (F(1, 216) = 160.95, *p* < 0.001). Post hoc comparisons showed that the knowledge increase over time was significant in Study 1 (*M*(T1) = 27.9, 95% CI [27.1; 28.6], *M*(T2) = 30.4, 95% CI [29.6; 31.2], (*t*(218) = 6.91, *p* < 0.001) as well as in Study 2 (*M*(T1) = 26.7, 95% CI [26.2; 27.3], *M*(T2) = 30.0, 95% CI [29.4; 30.6], (*t*(218) = 11.91, *p* < 0.001).

### 3.3. Hypotheses 3 and 4: Comparison to Control Condition

Finally, as Study 2 included an additional control condition with repeated response to the knowledge test without webinar participation in between measurements, we investigated whether the knowledge increase observed in the intervention condition did not occur in the control condition. This would indicate that the increase in the intervention condition was likely due to webinar participation and not due to practice effects by taking the knowledge test twice. Therefore, we hypothesized that in the matched sample of Study 2 (*n* = 126), there would be no differences between the control condition and the intervention condition at T1 (baseline; H3), and that the knowledge increase between T1 and T2 detected in the intervention condition would be smaller or non-significant in the control condition (H4). We conducted a Linear Mixed Model with Time (T1, T2) and Condition (Control, Intervention) as predictors. Predictors were entered as fixed effects and we included a random intercept for participants to account for the repeated measurement of the knowledge test. Results showed that there was a main effect of Condition (*F*(1, 126) = 7.81, *p* = 0.006) and a main effect of Time (*F*(1, 126) = 47.58, *p* < 0.001), which were qualified by the hypothesized two-way interaction of Condition and Time (*F*(1, 126) = 13.10, *p* < 0.001). In line with H3, the average baseline knowledge of participants in the control condition and the intervention condition did not significantly differ (*t*(172) = 1.16, *p* = 0.652). Moreover, in line with H4, post hoc comparisons showed that in the control condition, mean performance in the knowledge test was not higher at T2 compared to T1 (*M*(T1) = 27.1, 95% CI [26.1; 28.1]; *M*(T2) = 27.9, 95% CI [26.9; 28.9], *t*(128) = 2.09, *p* = 0.161). In contrast and as hypothesized, in the intervention condition, performance in the knowledge test was higher at T2 compared to T1 (*M*(T1) = 27.9, 95% CI [27.1; 28.6]; *M*(T2) = 30.4, 95% CI [29.6; 31.2], *t*(128) = 8.28, *p* < 0.001) (see Figure 1).

### 3.4. Exploratory Analyses: Investigation of Single Items of the Knowledge Test

Finally, an exploratory analysis took a closer look at the effects of the webinar by investigating performance differences on the separate items of the test. To do so, we computed paired-samples *t*-tests with Bonferroni-adjusted *p*-values for the three subsamples separately (Study 1, Study 2 control condition, Study 2 intervention condition). Results are depicted in Table 2 (right side). In the control condition of Study 2, most comparisons were non-significant (except for higher mean values at T2 compared to T1 on the items “One can die from ME/CFS” (true/false) and “ME/CFS is mainly diagnosed with” (a careful history according to established criteria/a physical examination/a psychiatric history)).

In contrast, in the intervention conditions of both studies, participants showed higher knowledge after the webinar as compared to their baseline knowledge on the majority of items. In all cases of significant comparisons, knowledge at T2 was higher than at T1. Most non-significant comparisons were due to ceiling effects (i.e., an item was answered correctly by more than 90% of participants).

## 4. Discussion

The current research picked up two recent developments during the COVID-19 pandemic: an increase in cases of ME/CFS and a shift from face-to-face to virtual CME. Building on these recent developments, we investigated whether a webinar as a web-based form of CME can increase German-speaking health professionals’ knowledge about ME/CFS. First, results showed that despite the fact that participants at baseline reported having experience with patients with ME/CFS and half of the sample had previously received some form of education about ME/CFS, confidence in diagnosing and treating the illness was only moderate-to-low. This pattern of results underscores the need to further educate health professionals about ME/CFS in order to improve patients’ medical care situation. Second, knowledge about the general epidemiology, definitions and clinical understanding, diagnostic criteria, as well as treatment of ME/CFS was higher after webinar participation compared to before. This knowledge increase did not occur in a waiting-list control condition, in which participants responded to the knowledge test twice without webinar participation between measurements. We are, thus, confident that the knowledge increase can be attributed to webinar participation and was not due to practice effects on the knowledge test.

It should be noted as well that the content of the webinars in Studies 1 and 2 was largely overlapping, but not exactly the same. Nevertheless, in both studies combined and separately, we were able to show a knowledge increase after webinar participation on a knowledge test that included quite broad questions about ME/CFS. This makes it likely that our results would be generalizable to other webinars focusing on a basic introduction to ME/CFS. However, we would like to emphasize that we put great importance on the fact that talks were based on current state-of-the art research about ME/CFS as a somatic illness and given by researchers and clinicians with high level of expertise and experience with patients with ME/CFS. It is likely that this experience and focus on evidence-based medicine contributed to the webinars’ effectiveness in increasing knowledge about ME/CFS. It should further be noted that despite the overall effects of the webinar, knowledge on two specific items did not increase in the intervention condition. The first item tapped into the distinction between chronic fatigue and ME/CFS. We adapted the item’s wording from the original study (original: “Myalgic Encephalomyelitis, Chronic Fatigue Syndrome and Post Viral Fatigue Syndrome all mean the same thing” (false); adapted: “ME/CFS and chronic fatigue are different” (true)). We did so because recent studies showed partial overlap of ME/CFS and post-COVID syndrome as a form of post-viral fatigue syndrome (e.g., [13]) and, therefore, the original item is now ambiguous. The adapted version aimed at tapping into the differentiation between the full clinical picture of ME/CFS (where chronic fatigue is only one of many symptoms) and chronic fatigue, which can also be a symptom of other illnesses like cancer or multiple sclerosis. However, in the adapted version, one third of participants were not able to differentiate between chronic fatigue and the full clinical picture of ME/CFS. Therefore, future education about ME/CFS should put greater emphasis on this distinction. The second item was developed by the project team and not included in the original study. It asked about a viral infection or a psychological trauma as possible triggers for ME/CFS. About one third of participants incorrectly indicated that a psychological trauma can trigger ME/CFS, which might reflect subtle psychologization of the illness—even though the vast majority of participants correctly indicated that ME/CFS is not a psychological/psychosomatic illness. Consequently, future education on ME/CFS should include a more fine-grained discussion of the harmful misconception of ME/CFS as psychosomatic.

### Strengths and Limitations

We performed risk-of-bias analyses recommended for intervention studies [43]. The strengths of the current study were that the study design did not pose a risk of bias, as it included an intervention and a control group (Study 2), as well as a cohort design (multiple assessments of the same participants) with pre-post intervention data. Concerning participant representativeness, the study included a random assignment of participants to the intervention or control condition. Therefore, the current study can be classified as a randomized controlled trial. However, further criteria to determine risk of bias for participant representativeness were not met by the current study. First, the follow-up rate (rate of participation in both T1 and T2) did not reach the recommended level of 80%, but only 70% of the initial baseline sample also participated in T2. Another limitation was that participants were not randomly selected from the population of health professionals in German-speaking countries. Health professionals actively registered for the webinar and, therefore, our sample only included participants who were already interested in ME/CFS and felt the need to receive further education about the illness. In contrast, Hng et al. [25] collected a sample of hospital doctors at an event with mandatory attendance. This also explains why the baseline knowledge in the current research was considerably higher than in the original study. Nevertheless, even in the current selective sample with comparatively high baseline knowledge about ME/CFS, the webinar still led to a significant knowledge increase. On some items, baseline knowledge was already very high (>90% of participants provided the correct response); thus, there was not much room for improvement due to ceiling effects. Consequently, future efforts to educate health professionals about ME/CFS should start earlier during (under)graduate medical studies (where only a small minority of the current sample reported to have received teaching about ME/CFS) and/or might be part of mandatory staff education in hospitals and other medical facilities. Lastly, concerning risk of bias for equivalence of comparison groups, the control group and the intervention group were equivalent at baseline on outcome measures, indicating that participants in both groups started the study at similar levels of knowledge about ME/CFS. There was no selective dropout between measurement points, and concerning sociodemographics, the comparison groups were equivalent at baseline on gender. However, participants in the control condition were significantly younger than participants in the intervention condition despite random assignment to conditions. It cannot be ruled out that there was selective dropout of older participants in the control condition.

Further strengths and limitations of the current research concern the form of CME as a live webinar. Following recent trends of digitalization due to the COVID-19 pandemic worldwide, we developed a CME event that was accessible for a high number of participants (up to 1000 per session) from different countries. The online format was cost-efficient and practical, as neither the experts nor the participants needed to travel. The webinar adhered to recommendations for webinars as CME developed during the pandemic, in that it was around two hours long, included several talks by experts in the field with a duration of max. 30 min each, took place outside of regular working hours on a weekday evening, included support in case of technical issues, allowed active participation via the chat function, was recorded and can be revisited at a later point in time on-demand, and was accredited as CME [40,42]. However, due to the high number of up to 1000 participants per webinar, no individualized teaching was possible and in the Q&A session, only a fraction of the posted questions could be answered by the experts. To complement the large-scale dissemination of basic knowledge about ME/CFS via webinars, further formats of continuing medical education should target smaller groups of health professionals who currently treat patients with ME/CFS and provide opportunities for case-based discussion and supervision. Finally, not all health professionals who attended the webinar also participated in the evaluation study. This is a common issue in evaluation research; however, the current study cannot rule out selection effects, as it was not possible to test for differences between participants and non-participants.

## 5. Conclusions

It is highly relevant and important to improve the medical care situation of patients with ME/CFS in Germany and around the world. The current study showed that a webinar of only 120–135 min duration is a cost-effective and efficient way to provide continuing medical education about ME/CFS to hundreds of health professionals at a time. Thereby, knowledge gaps about ME/CFS can be decreased—which is even more relevant since ME/CFS cases are on the rise after the onset of the COVID-19 pandemic.

## Figures and Tables

**Figure 1 healthcare-11-02186-f001:**
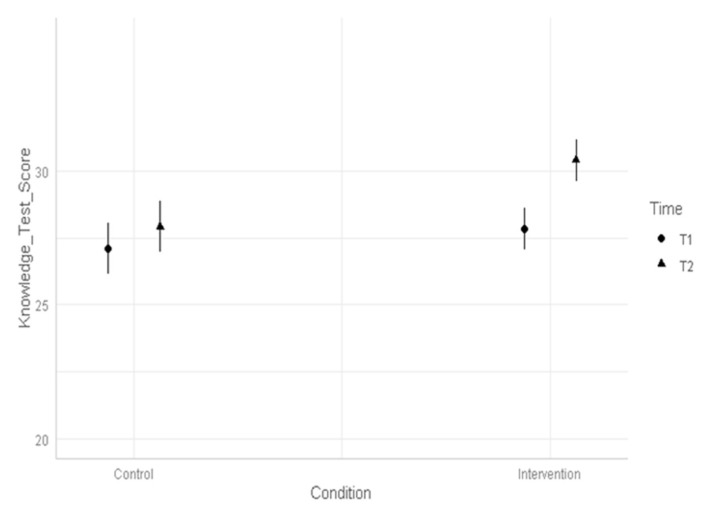
Mean knowledge test scores by Time and Condition.

**Table 1 healthcare-11-02186-t001:** Webinar content in Studies 1 and 2.

Fall 2021 (Study 1)	Fall 2022 (Study 2)
Overview about ME/CFS: Epidemiology, Pathogenesis, Diagnostics, and Treatment (30 min) ME/CFS and Post-COVID among Children and Adolescents (30 min) Research about Post-COVID (15 min) ME/CFS and Post-COVID in Private Practice (30 min) Q&A (30 min)	ME/CFS and Post-COVID: Definitions, Epidemiology and Diagnostics (20 min) ME/CFS and Post-COVID: Treatment and Pathogenesis (20 min) Inpatient Pain Therapy for ME/CFS (10 min) POTS and Autonomic Dysfunction (10 min) ME/CFS and Post-COVID in Private Practice (30 min) Q&A (30 min)

**Table 2 healthcare-11-02186-t002:** Items of the knowledge test and descriptive statistics for the subsamples.

	Study 1 Intervention Condition (N = 140)			Study 2 Control Condition (N = 50)			Study 2 Intervention Condition (N = 76)		
	T1	T2		T1	T2		T1	T2	
Item	*M* (*SD*)	*M* (*SD*)	*t (p)*	*M* (*SD*)	*M* (*SD*)	*t (p)*	*M* (*SD*)	*M* (*SD*)	*t (p)*
General Epidemiological Knowledge									
1. ME/CFS is rare Yes No (✓) completely correct response: 1 point	0.54 (0.50)	0.80 (0.40)	5.74 *** (<0.001)	0.50 (0.51)	0.56 (0.50)	1.00 (0.322)	0.53 (0.50)	0.72 (0.45)	3.33 ** (0.001)
2. ME/CFS affects more men women (✓) completely correct response: 1 point	0.96 (0.19)	0.99 (0.08)	2.02 * (0.045)	0.98 (0.14)	0.98 (0.14)	0.00 (1.00)	0.99 (0.11)	0.99 (0.11)	0.00 (1.00)
3. ME/CFS can affect children Yes (✓) No completely correct response: 1 point	0.96 (0.20)	1.00 (0.00)	2.49 * (0.014)	0.98 (0.14)	1.00 (0.00)	1.00 (0.322)	0.99 (0.11)	1.00 (0.00)	1.00 (0.321)
4. ME/CFS symptoms usually resolve within 6 months Yes No (✓) completely correct response: 1 point	0.93 (0.26)	0.91 (0.28)	0.53 (0.595)	0.82 (0.39)	0.86 (0.35)	0.81 (0.420)	0.93 (0.25)	0.91 (0.29)	0.70 (0.483)
5. ME/CFS is often painful True (✓) False completely correct response: 1 point	0.82 (0.38)	0.97 (0.17)	4.70 *** (<0.001)	0.80 (0.40)	0.86 (0.35)	1.14 (0.261)	0.86 (0.35)	0.97 (0.16)	3.17 ** (0.002)
6. ME/CFS often causes chronic disability True (✓) False completely correct response: 1 point	0.93 (0.26)	0.95 (0.22)	0.77 (0.441)	0.96 (0.20)	0.96 (0.20)	0.00 (1.00)	0.93 (0.25)	0.93 (0.25)	0.00 (1.00)
7. Children and adolescents with ME/CFS often miss longer periods of school True (✓) False completely correct response: 1 point	0.94 (0.25)	0.99 (0.08)	2.90 ** (0.004)	1.00 (0.00)	1.00 (0.00)	0.00 (1.00)	1.00 (0.00)	1.00 (0.00)	0.00 (1.00)
Definitions and Clinical Understanding									
8. ME/CFS is a psychological/ psychosomatic illness physical illness (✓) completely correct response: 1 point	0.84 (0.37)	0.96 (0.19)	4.38 *** (<0.001)	0.88 (0.33)	0.88 (0.33)	0.00 (1.00)	0.87 (0.34)	0.96 (0.20)	2.41 * (0.019)
9. ME/CFS and chronic fatigue are different things True (✓) False completely correct response: 1 point	0.66 (0.47)	0.66 (0.47)	0.00 (1.00)	0.72 (0.45)	0.64 (0.48)	1.27 (0.209)	0.70 (0.46)	0.71 (0.46)	0.21 (0.836)
10. ME/CFS can affect the cardiovascular system (✓) the musculoskeletal system (✓) the nervous system (✓) the immune system (✓) the endocrine system (✓) cellular metabolism (✓) the gastrointestinal system (✓) completely correct response: 7 points	5.84 (1.60)	6.49 (1.02)	5.20 *** (<0.001)	6.04 (1.38)	6.12 (1.32)	0.45 (0.655)	6.20 (1.24)	6.46 (1.03)	2.16 * (0.034)
11. One can die from ME/CFS True (✓) False completely correct response: 1 point	0.31 (0.47)	0.40 (0.49)	2.39 * (0.018)	0.28 (0.45)	0.44 (0.50)	2.68 * (0.010)	0.49 (0.50)	0.59 (0.49)	2.04 * (0.045)
Diagnostic Process and Diagnostic Criteria									
12. ME/CFS is mainly diagnosed with a careful history according to established criteria (✓) a physical examination (✓) a psychiatric history completely correct response: 3 points	1.66 (0.80)	2.16 (0.50)	7.46 *** (<0.001)	1.62 (0.78)	1.86 (0.67)	2.37 * (0.022)	1.83 (0.76)	2.09 (0.61)	2.85 ** (0.006)
13. The diagnosis of ME/CFS requires six months of substantial reduction in functioning with fatigue (✓) psychiatric symptoms signs of anxiety and/or depression worsening of symptoms after activity (post-exertional malaise) (✓) neurocognitive symptoms (✓) disordered sleep (✓) completely correct response: 6 points	4.99 (1.00)	5.26 (0.84)	3.25 ** (0.001)	5.00 (0.88)	5.10 (0.97)	0.93 (0.358)	4.93 (1.00)	5.37 (0.85)	3.79 *** (<0.001)
14. Sensible diagnosis for ME/CFS patients can be a Schellong test (✓) a hand strength test (✓) measurement of LDH (✓) completely correct response: 3 points	2.24 (0.43)	2.39 (0.49)	3.71 *** (<0.001)	2.12 (0.44)	2.28 (0.45)	1.74 (0.088)	2.28 (0.45)	2.39 (0.49)	2.00 * (0.049)
15. ME/CFS can be triggered by a viral infection (✓) a psychological trauma completely correct response: 2 points	1.66 (0.51)	1.56 (0.50)	1.89 (0.061)	1.72 (0.45)	1.66 (0.48)	1.00 (0.322)	1.62 (0.54)	1.74 (0.44)	1.91 (0.060)
Treatment									
16. ME/CFS can be treated with antiviral medication (✓) activation therapy (Graded Exercise Therapy, GET) vitamin supplements (✓) cognitive behavioral therapy completely correct response: 4 points	1.48 (1.27)	2.49 (1.17)	9.13 *** (<0.001)	1.72 (1.21)	1.72 (1.16)	0.00 (1.00)	1.75 (1.23)	2.58 (0.93)	6.64 *** (<0.001)
17. Patients should be advised to stay within their energy levels and not to overextend themselves (pacing). True (✓) False completely correct response: 1 point	0.98 (0.15)	1.00 (0.00)	1.75 (0.083)	0.98 (0.14)	1.00 (0.00)	1.00 (0.322)	0.97 (0.16)	1.00 (0.00)	1.42 (0.159)

Notes. Correct response options are indicated by checkmarks. Participants received one point for each correct response (correct option checked, incorrect option not checked). The maximum possible number of points for completely correct responses was 36. * *p* < 0.05, ** *p* < 0.01, *** *p* < 0.001.

## Data Availability

Data are provided on the Open Science Framework (OSF; https://osf.io/8m32c).

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
