# Peer review of "Evaluation of a Webinar to Increase Health Professionals’ Knowledge about Myalgic Encephalomyelitis/Chronic Fatigue Syndrome (ME/CFS)"

_healthcare, 2023, doi:10.3390/healthcare11152186_

Round 1

Reviewer 1 Report

Thank you for having the chance to review this interesting study. I hope this study will make a contribution toward improving health professionals’ knowledge.

I think that this manuscript needs some revision. Please check out the following:

Keywords: I think that the keywords are drawn from the MeSH list. But “ME/CFS” and “Webinars” were not on the MeSH list. “Chronic Fatigue Syndrome” is the entry terminology for “Fatigue Syndrome, Chronic”. In addition, “Continuing Medical Education” is the entry terminology for “Education, Medical, Continuing”.

I think you had better write “Fatigue Syndrome, Chronic” and “Education, Medical, Continuing”.

2.1. Participants

Line 165-171: The estimated sample sizes were achieved in both studies (Study 1: T2: n=145; Study 2: T2: n=128)... Participants were excluded from the analyses (Study 1: n=4, Study 2: n=2)... To investigate H2-H4, we combined the matched T1-T2 samples of both studies, the total sample size of the combined dataset was N=266.

Is it the right number? Isn’t it N=267? Please check it out.

4. Discussion

The first paragraphs are already written in the part about the introduction. I think you had better avoid repetition.

Give a cautious overall interpretation of the results, considering results from similar studies and the references. 

Reviewer 2 Report

The overall English quality was sound.

Round 2

Reviewer 2 Report

Thank you for resubmitting the manuscript and addressing the comments of the reviewer. The reviewer has confirmed that the authors have addressed all of the comments of the reviewer. Although the reviewer is mostly satisfied with the revisions and comments provided by the authors, the reviewer would like to present several points that should be addressed further. 

Firstly, thank you for adding additional information about CME in the introduction. Although this information is relevant to the manuscript, and in the personal opinion of the reviewer, this information should remain in the manuscript, the reviewer's intention was to have authors add further information on webinars hosted worldwide during the pandemic. Although the new information added by the authors are important and informative to the readers, this specific information focused on Germany and Austria makes the impression that this manuscript is centralized to these two countries. The reviewer believes that the findings of this manuscript would be feasible to other countries as well, and therefore believe that information regarding to webinars from other countries should be added as well, especially in the introduction or discussion section. 

Secondly, the authors should reconsider the location of the information they provide, especially in the M and R of the IMRAD structure. For example, the phrase "table 2" should not appear in the methods section 2.3, as this table is the "result" of such methods/procedures. Such presentation of information will confuse the readers (the readers would think that table 2 would be presented close by). Although the reviewer chose table 2 as an example of such features as table 2 was discussed in a different comment in the previous review process, this is not the only information that has the potential of confusing the readers. Please check once again if the information provided in the manuscript are adequate. 

Thank you for the opportunity to comment again on this manuscript. I hope that the authors would find these comments useful. 
